# Monitoring SARS-CoV-2 seroprevalence over time among pregnant women admitted to delivery units: Suitability for surveillance

**Mariana Yumi Miyadahira**[1], **Maria de Lourdes Brizot**[1], **Neal Alexander**[2], **Ester Cerdeira Sabino**[3], **Lea Campos de Oliveira da Silva**[4], **Mara Sandra Hoshida**[1,5], **Ana Maria da Silva Sousa Oliveira**[1], **Ana Claudia Silva Farche**[1], **Rossana Pulcineli Vieira Francisco**[1,6], **Philippe Mayaud**[7] *

**1** Disciplina de Obstetrícia, Departamento de Obstetrícia e Ginecologia da Faculdade de Medicina da Universidade de São Paulo (FMUSP), São Paulo, Brazil, **2** Faculty of Epidemiology & Population Health, London School of Hygiene & Tropical Medicine, London, United Kingdom, **3** Departamento de Moléstias Infecciosas e Parasitárias, Instituto de Medicina Tropical da FMUSP, São Paulo, Brazil, **4** Laboratório de Medicina Laboratorial—LIM 03, Hospital das Clínicas da FMUSP, São Paulo, Brazil, **5** Laboratório de Investigação Médica- LIM 57, Hospital das Clínicas da FMUSP, São Paulo, Brazil, **6** Hospital Universitário da Universidade de São Paulo (USP), São Paulo, Brazil, **7** Faculty of Infectious & Tropical Diseases, London School of Hygiene & Tropical Medicine, London, United Kingdom

* Philippe.Mayaud@lshtm.ac.uk

**Data Availability Statement:** Data have been made available at https://osf.io/8QHZP/.

## Abstract

### Objectives

To determine SARS-CoV-2 seroprevalence over time and risk factors among pregnant women at delivery in São Paulo, Brazil; and to evaluate the suitability of pregnant women as a sentinel population for SARS-CoV-2 serosurveillance.

### Methods

Unselected consecutive pregnant women presenting at the labor ward of a single large hospital between July 20th 2020 to February 21st 2021 were enrolled and tested for SARS-CoV-2 serology using two assays: the rapid chromatic Wondfo One Step (for total IgA and IgG detection) and Roche Elecsys assay (detecting anti-nucleoprotein [N] IgG). SARS-CoV-2 seroprevalence was computed as smooth spline function over time with 95% confidence intervals (CI). Risk factors were evaluated for positivity by each assay. We compared time-point seroprevalence by the two assays with four concomitant community household surveys (HHS), in which the Roche assay was used, to determine the sensitivity and relevance of the pregnant women population as sentinel population.

### Results

Overall SARS-CoV-2 seroprevalence was 28.9% (221/763) by Roche and 17.9% (137/763) by Wondfo. Reported symptoms experienced during pregnancy were all significantly correlated with being SARS-CoV-2 seropositive at delivery with any assay (with odds-ratios ranging from 3.0 [95% CI: 2.1–4.3] for coryza to 22.8 [95% CI: 12.3–46.6] for ageusia). Seropositivity by either assay was high in women at delivery in the early period of the

**Funding:** Funding was provided by CAPES-Coordination of Superior Level Staff Improvement (88881.504727/2020-01), for Universidade de São Paulo (Prof. Rossana Pulcineli Vieira Francisco) and by the European Union's Horizon 2020 Research and Innovation Programme under ZIKAlliance Grant Agreement No 734548, for the London School of Hygiene & Tropical Medicine contract (LSHTM PI: Prof Philippe Mayaud). The funders had no role in the study design, data collection and analysis, decision to publish, or preparation of the manuscript.

**Competing interests:** The authors have declared that no competing interests exist.

pandemic (June 2020), compared with seropositivity in women from the concomitant HHS: 44.1% (95% CI: 21.8–66.4) for Roche, 54.1% (30.9–78.5) for Wondfo, versus 11.4% (95% CI: 9.2–13.6) for HHS. For later periods (October 2020 and January 2021), the seropositivity in women at delivery measured by Roche corresponded well with the prevalence found among women in the HHS using the same assay, whilst prevalence measured by Wondfo dropped.

## Conclusions

Women at delivery represent a highly exposed and readily accessible population for sentinel surveillance of emerging infections such as SARS-CoV-2.

## Introduction

COVID-19 (coronavirus disease 2019) caused by SARS-CoV-2 (severe acute respiratory syndrome coronavirus 2), a virus completely unknown to the world before the end of 2019, was declared a global pandemic by the World Health Organization (WHO) in March 2020. By May 31st 2022, there had been over 6.0 million deaths worldwide [1], including over 665,000 in Brazil [2], and over 42,000 in the city of São Paulo alone [3]. Whilst risk factors for severe COVID-19 are increasingly known and include age, sex, comorbidities and socio-economic factors [4–8], those for acquiring and transmitting SARS-CoV-2 infection are less well identified.

Physiological changes in the respiratory and immune systems may make pregnant women more susceptible to infection [9]. Recent data suggest that pregnant women might be at increased risk of severe COVID-19. Studies have shown that compared to non-pregnant women, pregnant women are significantly more likely to be hospitalized, admitted to intensive care units (ICU), receive invasive ventilation or extracorporeal membrane oxygenation (ECMO), and die [10–16]. In addition, results from a multinational cohort study pointed out a higher rate of pregnancy related complications, such as hypertensive disorders of pregnancy and fetal distress [17]. There is evidence of significantly elevated risk of preterm birth [13–17] and lower neonatal weight [17] among SARS-CoV-2 affected pregnancies when compared with pregnant women without SARS-CoV-2 infection, even in milder disease. Also, the neonates who tested positive (born to COVID-19 positive mothers) had worse outcomes, with higher rates of ICU admission, fever, gastrointestinal and respiratory symptoms and death, even after adjusting for prematurity [17]. Therefore, this complicated disease course seems to compromise the development and survival of the fetus and neonate, although robust, organized, homogeneous information on the effect of COVID-19 on perinatal, neonatal and long-term outcomes remain insufficient [18–21].

Universal testing for SARS-CoV-2 infection in all pregnant women admitted at labor and delivery (L&D) units has been implemented in a range of settings [22–27]. Screening can be done by virus antigen detection, mostly relying on molecular methods such as reverse-transcriptase PCR, or by serological testing. The first approach will verify the immediate risk of infection in the mother and potential risk of perinatal transmission. Additionally, antigen-detection screening can be useful to organize hospital beds and guide services' isolation practices, neonatal care and the use of personal protective equipment [28], but will likely under-evaluate the risk associated with virus acquisition during pregnancy, since this test detects only transient acute infection [29].

Conversely, in a previously unexposed (and unvaccinated) population of pregnant women, serological status provides information on cumulative exposure to the virus over the duration of pregnancy (assuming antibody decay following natural infection is not much inferior to pregnancy duration) and its possible effects on pregnancy outcome, frequency of asymptomatic and pauci-symptomatic infections [30–33].

Besides verifying exposure of the mother (and consequently of the fetus), and therefore, guiding management, this strategy represents a great opportunity to obtain estimates of SARS-CoV-2 circulation in the community [34]. Also, pregnant women, particularly at the time of delivery, might be a good sentinel population for surveillance of the infection during restrictive quarantine times, when these women must still interact with the health system, whilst most others can avoid or defer doing so.

In the present study we sought to determine the seroprevalence and risk factors of SARS-CoV-2 among pregnant women admitted in L&D units in a public maternity hospital complex in São Paulo, during the first year of the SARS-CoV-2 pandemic in Brazil; and to evaluate the suitability of this population in serosurveillance, contrasting with data obtained from serial community household surveys conducted concomitantly in the city [35].

## Materials and methods

### Settings and populations

From April 12[th] 2020, soon after the introduction of SARS-CoV-2 in São Paulo [36], we have been conducting a prospective evaluation of SARS-CoV-2 in all pregnant women presenting for delivery in labor and delivery (L&D) units of a large public hospital complex. Details of the study have been described elsewhere [21]. In brief, the hospital complex consists of two separate L&D units in Hospital das Clínicas and in Hospital Universitário, which are interconnected via cross-referral systems for complex pregnancy cases in usual times. Under newly established COVID-19 management structures, Hospital das Clínicas was the referral hospital for known COVID-19 cases (for all cases, including pregnant women) and Hospital Universitário was a non-COVID-19 hospital.

For the present investigation, consecutive pregnant women admitted for delivery at the L&D unit at Hospital Universitário from July 20[th] 2020 to February 21[st] 2021, were invited to participate. The research team was present for 8 hours each weekday. Those pregnant women of all ages who provided signed written informed consent to participate in the study were eligible and included.

Data on participants' socio-demographics, clinical and obstetrical history, details of previous treatments, exam results (including serologies for HIV, syphilis, hepatitis B, hepatitis C, toxoplasmosis, rubella, cytomegalovirus), hospital admissions during pregnancy, experience of COVID-19 or flu-like symptoms during pregnancy were obtained and recorded in a REDCap platform database (Research Electronic Data Capture Consortium, supported by the National Institutes of Health, USA). The following symptoms were considered indicative of COVID-19 according to the WHO case definition at the time: fever, cough, sore throat, coryza, dyspnea, headache, myalgia, asthenia, fatigue, diarrhea, anosmia and ageusia [37]. Delivery and neonatal data were retrieved from patients and their newborns' medical records, immediately after delivery. Medical records were reassessed to get patients´ dates of hospital release and outcome. These data were also recorded in a REDCap database. Maternal venous blood samples were collected at admission for serological testing.

Seroprevalence among pregnant women over time was compared to four consecutive rounds of household surveys (HHS) (each phase comprised 9 to 10 days) that were conducted by the Research and Development Sector of Grupo Fleury (a large private laboratory in São

Paulo), which took place over the same study period. The HHS was conducted to monitor the seroprevalence of SARS-CoV-2 infection in adults in the city of São Paulo during the pandemic development [35]. In brief, the population was sampled using a probabilistic two-stage method, census sector and household. All residents in the selected households over the age of 18 years old were invited to participate. Two specific survey strata were created, city districts with the highest and lowest average household income [35], each corresponding to about half of the surveyed population, and data was stratified by sex. Following administration of a face-to-face questionnaire, a venipuncture blood sample was drawn from each participant. We excluded data from the first (pilot) phase of the HHS conducted prior to our own survey.

Data are presented according to STROBE (Strengthening the reporting of observational studies in epidemiology) recommendations [38]. The checklist is included as S1 Table.

## Laboratory methods

**SARS-CoV-2 serology.** For serological screening at delivery, a qualitative rapid chromatographic immunoassay (RCI) (measuring IgG and IgM antibodies against the SARS-CoV-2 S1 protein) was performed using the Wondfo One Step COVID-19 test (Guangzhou Wondfo Biotech, China) as previously described [39]. Results were read within 15 minutes by three appropriately trained people and communicated back to the L&D unit within 24 hours of collection.

Subsequently, all stored samples were retested in the same laboratory with the Elecsys anti-SARS-CoV-2 E2G300 (Roche Diagnostics, Rotkreuz, Switzerland), an electrochemiluminescence assay that allows a qualitative detection of specific SARS-CoV-2 antibodies. The assay uses a recombinant protein from the nucleocapsid (NC) antigen of the virus with the double antigen sandwich methodology, which favors the detection of high affinity antibodies against SARS-CoV-2.

For the household surveys, serological initial testing measured the presence of anti-SARS-CoV-2 IgM and IgG antibodies using a chemiluminescence assay in survey phases 1 and 2 (Maglumi chemiluminescence assay, Snibe [Shenzhen New Industries Biomedical Engineering Co., Ltd.], Shenzhen, China) (May and June 2020). In later phases 3 (July 2020), 4 (October 2020) and 5 (January 2021), the Roche Elecsys assay was performed in combination with the Maglumi assay. A sample was considered seropositive if any of the two performed tests had a positive result [35].

**Additional SARS-CoV-2 testing.** A subgroup of women was tested during pregnancy in their own hospital or in private laboratories either by molecular tests of naso-pharyngeal swabs or by serology, because of their job requirements, presence of symptoms or suspicion of being a contact. The results of these tests were evidenced by the printed copy showed by the patient at the L&D admission.

## Statistical analysis

Descriptive statistics (frequencies) were calculated including 95% confidence intervals (CI) for binomial proportions using the Wilson method [40]. Positivity on the Wondfo was compared with that on Roche by McNemar's test and the 95% CI for difference in proportions [41]. Cohen's $\kappa$ (kappa) was also calculated [42]. Values of $\kappa$ were categorized following Landis & Koch [43].

Risk factors for seropositive status at delivery, as determined by the Roche assay, were analyzed in terms of odds ratios (OR) with their 95% CI and *p* values from Wald standard errors in logistic regression. For this regression analysis, each independent variable was analyzed

separately, those with missing values for that variable were excluded, and height and body mass index (BMI) were each included as a continuous variable.

Changes over time were described for each diagnostic test by using polynomial splines with B-spline basis [44], and 5 degrees of freedom, to estimate the seroprevalence as a smooth function of epidemiological week, again with logistic regression. Seroprevalence in phase 2 of the HHS was compared descriptively with delivery seroprevalence in the calendar month of June 2020: for phases 3 to 5, the months were July 2020, October 2020 and January 2021 respectively. For each phase, the fitted value from the spline was taken from the epidemiological week closest to the midpoint of the month in question. Analyses were carried out using R software version 4.1.0 was used, with packages "splines", "binom", "irr" and "exact2x2".

## Ethical considerations

The study protocol was approved by the ethics committees of both hospitals, Hospital das Clínicas and Hospital Universitário (CAAE: 30270820.3.0000.0068, approved April 11[th], 2020 and CAAE: 30270820.3.3001.0076, approved May 20[th] 2020), and was registered at Clinical-Trial.gov (NCT04647994). All participants provided signed written informed consent.

## Results

### Population characteristics

A total of 1,867 unselected consecutive pregnant women were admitted for delivery or curettage at Hospital Universitario in São Paulo and were eligible to participate in the study over an eight-month period (June 20[th], 2020–February 21[st], 2021) (i.e., about 233 women per month), of whom 997 (53.4%) were admitted during periods when it was not possible to include them in the study for logistical reasons (i.e., night shifts and weekends). Of the remaining 870 women, 98 (11.3%) declined participation, leaving 772 eligible and consenting women participating in the study (Fig 1).

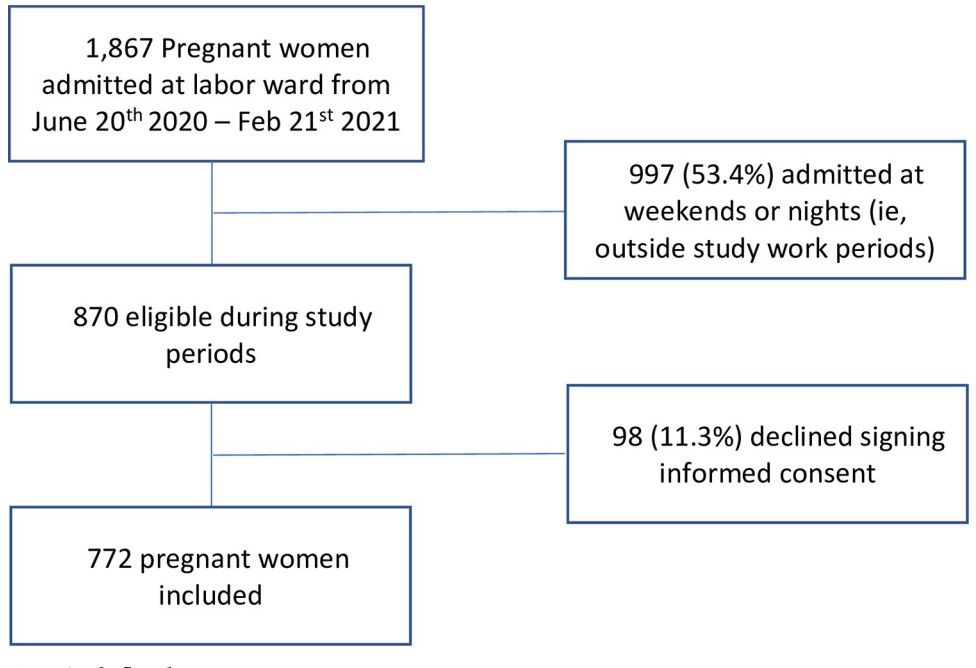

**Fig 1. Study flowchart.**

**Table 1. Population characteristics.** (N = 772 participants).

| | Frequency n/N (%) or mean (±SD) or median (IQR) |
|---|---|
| Maternal age (years, N = 770) | 29.1 (± 6.6) |
| Ethnicity | |
| White | 287/769 (37.3) |
| Non-white | 482/769 (62.7) |
| BMI group | |
| $< 30$ kg/m$^2$ | 345/746 (46.2) |
| $\geq 30$ kg/m$^2$ | 401/746 (53.8) |
| BMI (kg/m$^2$, N = 746) | 31.3 (± 5.9) |
| Height (cm, N = 762) | 160.8 (± 6.5) |
| HDI (scaled to max. value 100; N = 746), median (IQR) | 77.6 (67.8–80.5) |
| Years of schooling (N = 766) | 11.80 (± 2.73) |
| Occupation | |
| Healthcare workers | 52/758 (6.9) |
| Essential non-healthcare workers[a] | 92/758 (12.1) |
| Other (non-essential) workers | 614/758 (81.0) |
| Smoking | 72/770 (9.4) |
| Alcohol use | 56/759 (7.4) |
| Comorbidities | |
| Diabetes | 21/767 (2.7) |
| Hypertension | 41/767 (5.3) |
| Cardiac | 19/768 (2.5) |
| Lung | 49/768 (6.4) |
| Rheumatic | 19/768 (2.5) |
| Cancer | 6/772 (0.8) |
| Thromboembolic | 10/772 (1.3) |
| Any one of the above comorbidities | 147/767 (19.2) |
| Nulliparous | 315/771 (40.9) |
| HIV seropositive | 8/736 (1.1) |
| Syphilis seropositive[b] | 35/722 (4.8) |
| Hepatitis C seropositive | 3/428 (0.7) |
| Hepatitis B (anti-HBs) seropositive | 105/646 (16.3) |

BMI = body mass index; HDI = human development index

Where the denominator of a proportion differs from the overall sample size of 772, this is because of missing data.

[a] Essential workers defined as those working with food supply or hospital services such as cleaning and hospitality

[b] Either positive by treponemal assay (e.g., Treponema pallidum particle assay [TPPA]) in those who had never tested positive for syphilis) and/or by non-treponemal assay (e.g., Venereal Diseases Research Laboratory [VDRL]) with a positivity cut-off titer of $\geq$1:4

All participants were initially screened with the rapid SARS-CoV-2 serological assay (Wondfo) to identify potentially exposed women. There was sufficient leftover sample to test 763 (99%) participants with the Roche assay allowing for the comparative study of seroprevalence over time with the community household surveys.

Table 1 shows participants' characteristics. The mean maternal age was 29.1 years old; 41% were nulliparous; over half had a body mass index (BMI) $\geq$30 kg/m$^2$ (defined as obese); at least one chronic comorbidity was observed in 19% of participants, with lung disease the most

prevalent condition (6.4%); 4.9% had reactive syphilis serology (either positive by treponemal assay [e.g., TPPA, *Treponema pallidum* particle agglutination assay] in those who had never tested positive for syphilis and/or positive by a non-treponemal assay [e.g, Venereal Diseases Research Laboratory, VDRL] at a cut-off titer of $\geq$1:4), which was detected and treated during antenatal care; and 1% were HIV seropositive, all of whom were on antiretroviral therapy (ART) at the time of delivery.

Overall, 32 (4%) women attended for curettage following incomplete miscarriage. Among 738 (96%) women attending for delivery, the median gestational age was 39 weeks, there were 16 (2%) twin and 2 (0.2%) triplet pregnancies, and over half of the women (55.7%) were delivered by cesarean section. Nearly all pregnancies resulted in livebirths (99.8%) with a median birthweight of live neonates of 3,230 g (interquartile range, 2,840–3,575). Overall, 78 neonates (10.5%) were born with neonatal malformations, as high-risk pregnancies from Hospital das Clínicas were referred to Hospital Universitario for delivery in the absence of acute COVID-19 symptoms, according to the institutional COVID-19 protocol.

## SARS-CoV-2 seroprevalence and risk factors

The overall seroprevalence was 17.9% (137/763) by Wondfo and 28.9% (221/763) by Roche, and 30.3% (231/763) by either assay. Relative to Roche, the Wondfo test therefore categorized 11.0% fewer positive (95% CI: 8.3–13.7, p<0.001). The results of serological testing by the two assays showed an overlap of 127 cases detected by both assays (ie, 55% of all 231 positive by either assay). There were 10 Wondfo positive results (7.3%, 10/137) which were not positive by Roche. Overall, 532/763 (69.7%) samples were negative by both assays. Assay agreement was "substantial" at $\kappa$ = 0.63 [44].

Over a third of women reported symptoms indicative of possible SARS-CoV-2 infection/ COVID-19 (35.7%, 273/763) during their pregnancy. For each assay, about 60% of positive results were found among symptomatic women. The Roche assay identified the greatest proportion of SARS-CoV-2 seropositive women among both symptomatic (46.9%, 128/273) and asymptomatic women (19.0%, 93/490), compared to Wondfo (28.6%, 78/273; and 12.0%, 59/490, both p<0.001). In addition, 50 (asymptomatic and symptomatic) women with positive SARS-CoV-2 nasopharyngeal swab during pregnancy had results from both the Roche and Wondfo assays. Among these, the Roche assay identified more than the Wondfo assay: 46/50 (92.0%) as opposed to 31/50 (62.0%), a difference of 30.0% (95% confidence interval 13.5–44.6%, p<0.001).

We conducted separate risk factor analyses for seropositivity by each of the assays. Given that Roche seemed to be the most sensitive assay, we present population characteristics and risk factor analyses based on the Roche serological results in Table 2. Similar analyses with the Wondfo assay are presented in S2 Table.

Overall, 21 variables were included in the risk factor analysis. Factors associated with SARS-CoV-2 seropositivity and other selected variables are presented in Table 2.

Seropositive pregnant women had lower height (OR = 0.96 per cm, 95% CI: 0.93–0.98) than seronegative women. Reported COVID-19 symptoms during pregnancy (fever, cough, sore throat, myalgia, asthenia, coryza, anosmia, ageusia, dyspnea, headache and fatigue) were all significantly more prevalent in seropositive pregnant women (with odds ratios ranging from 3.0 for coryza to 22.8 for ageusia), whilst the lack of symptom reporting was inversely correlated with seropositivity (42.1% of asymptomatic pregnant women during pregnancy were SARS-CoV-2 seropositive). Associations were stronger for symptoms now known to be more closely associated with the first wave of COVID-19, such as ageusia, anosmia, dyspnea, myalgia and fatigue.

**Table 2. Factors associated with SARS-CoV-2 seropositivity by Roche.**

| | Total N = 763 | SARS-CoV-2 seropositive N = 221 (29%) | SARS-CoV-2 seronegative N = 542 | OR (95%CI) | p-value |
|---|---|---|---|---|---|
| **Age** | | | | | |
| <30 years | 409 (53.6) | 129 (58.4) | 280 (51.7) | 1 | |
| ≥30 years | 354 (46.4) | 92 (41.6) | 262 (48.3) | 0.8 (0.6–1.0) | 0.092 |
| **Ethnicity** | | | | | |
| White | 283 (37.2) | 80 (36.4) | 203 (37.6) | 1 | |
| Non-white | 477 (62.8) | 140 (63.6) | 337 (62.4) | 1.1 (0.8–1.5) | 0.751 |
| **Occupation** | | | | | |
| Healthcare | 52 (6.9) | 17 (8.0) | 35 (6.5) | 1 | |
| Essential non-healthcare[a] | 90 (12.0) | 30 (14.1) | 60 (11.2) | 1.0 (0.5–2.2) | 0.938 |
| Non-essential | 607 (81.0) | 166 (77.9) | 441 (82.3) | 0.8 (0.4–1.5) | 0.410 |
| **Any comorbidity[b]** | | | | | |
| No | 613 (80.3) | 179 (81.0) | 434 (80.1) | 1 | |
| Yes | 150 (19.7) | 42 (19.0) | 108 (19.9) | 0.9 (0.6–1.4) | 0.771 |
| **BMI** (kg/m$^2$) | 31.3 (5.9) | 32.1 (6.2) | 31 (5.8) | 1.03 (1.00–1.06) | 0.024 |
| **Height** (cm) | 160.8 (6.5) | 159.5 (6.1) | 161.3 (6.6) | 0.96 (0.93–0.98) | 0.001 |
| **Potential COVID-19 symptoms during pregnancy** | | | | | |
| Fever | 78 (10.2) | 45 (20.4) | 33 (6.1) | 3.9 (2.4–6.4) | <0.001 |
| Cough | 134 (17.6) | 68 (30.8) | 66 (12.2) | 3.2 (2.2–4.7) | <0.001 |
| Sore throat | 80 (10.5) | 42 (19.0) | 38 (7.0) | 3.1 (1.9–5.0) | <0.001 |
| Myalgia | 98 (12.8) | 64 (29.0) | 34 (6.3) | 6.1 (3.9–9.7) | <0.001 |
| Asthenia | 89 (11.7) | 58 (26.2) | 31 (5.7) | 5.9 (3.7–9.5) | <0.001 |
| Coryza | 164 (21.5) | 79 (35.7) | 85 (15.7) | 3.0 (2.1–4.3) | <0.001 |
| Anosmia | 86 (11.3) | 74 (33.5) | 12 (2.2) | 22.2 (12.2–44.1) | <0.001 |
| Ageusia | 82 (10.7) | 71 (32.1) | 11 (2.0) | 22.8 (12.3–46.6) | <0.001 |
| Dyspnea | 72 (9.4) | 51 (23.1) | 21 (3.9) | 7.4 (4.4–13.0) | <0.001 |
| Headache | 137 (18.0) | 75 (33.9) | 62 (11.4) | 4.0 (2.7–5.9) | <0.001 |
| Fatigue | 82 (10.7) | 55 (24.9) | 27 (5.0) | 6.3 (3.9–10.5) | <0.001 |
| No symptoms | 490 (64.2) | 93 (42.1) | 397 (73.2) | 0.3 (0.2–0.4) | <0.001 |

Data shown as frequency, n/N (%) or mean (±SD)

BMI = body mass index; CI = confidence interval; OR = odds ratio

[a] Essential workers defined as those working with food supply or hospital services such as cleaning and hospitality

[b] Any comorbidity includes diabetes, hypertension, cardiac, lung, rheumatic diseases, cancer and thromboembolic events

## SARS-CoV-2 seroprevalence over time and comparison with household community surveys

Fig 2 shows SARS-CoV-2 seroprevalence over time among women at delivery, by each of the two assays. SARS-CoV-2 seropositivity was high in June 2020 by both assays, with a visible decline for the following months, with subsequent rise at the beginning of 2021, coinciding with the observed case load in the community [3].

Table 3 presents the results of SARS-CoV-2 seropositivity over time comparing results among pregnant women with both assays with results of the community household survey (HHS) at concomitant timepoints. Results of HHS are presented as overall prevalence, stratified by wealthiest and poorest strata (two strata representing 50% of respondents each), and for women only results. In phase 2 (June 2020) seroprevalence among women at delivery was higher than in the HHS population: women at delivery, 44.1% (21.8–66.4) by Roche, and

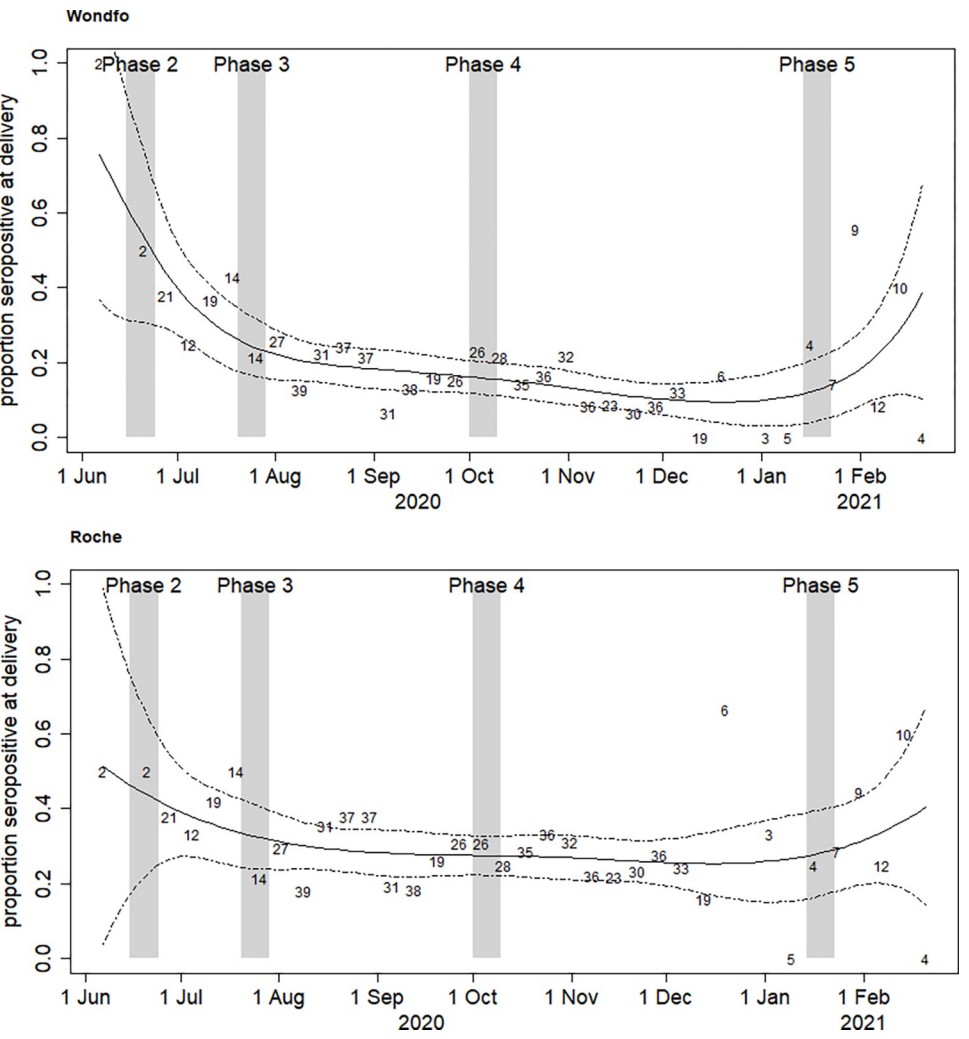

**Fig 2.** SARS-CoV-2 seroprevalence over time (June 2020-February 2021) among pregnant women at delivery in São Paulo, by Wondfo (Panel A) and Roche Elecsys (Panel B) assays. The grey bars show the dates of the second to fifth household surveys (HHS). The first survey is not shown because it was before the start of the study in pregnant women (see dates in Table 3). The numbers in the body of the figure are the numbers tested in each epidemiological week. The solid line is the estimate of the proportion seropositive from the spline fit, and the two dashed line are the 95% confidence limits.

54.1% (30.9–78.5) by Wondfo; compared with 11.2% (8.5–13.9) among women of HHS, and 16% (12.2–19.8) for the respondents from the poorest strata, all tested by Maglumi only. Interestingly, at this stage seropositivity by Wondfo was highest. In phase 3 (July 2020), this difference narrowed: women at delivery, 32.7% (24.1–41.4) by Roche; HHS women, 18.4% (15.2–22.1) by a combination of Roche and Maglumi, although by then, the seroprevalence measured by Wondfo was dropping among women at delivery to 24.7% (16.7–32.7). In phase 4 (October 2020) and phase 5 (January 2021), the seroprevalence in women at delivery by Roche (27.5% and 28.0%, respectively) corresponded well with the seroprevalence found among women in the HHS using the Roche and Maglumi combination (25.7% and 30.7% in women, respectively), whilst seroprevalence by Wondfo had dropped substantially (at 15.9% and 12.5% in Phases 4 and 5, respectively).

**Table 3. SARS-CoV-2 seroprevalence by different assays at selected periods corresponding to phases of community household surveys (HHS).**

| | SARS CoV-2 SEROPOSITIVE AT DELIVERY, % (95% CI) | | | | |
|---|---|---|---|---|---|
| | **PHASE 1** | **PHASE 2** | **PHASE 3** | **PHASE 4** | **PHASE 5** |
| | **May 4th - 12th** | **June 15th - 24th** | **July 20th - 29th** | **October 1st - 10th** | **January 14th -** |
| | **2020** | **2020** | **2020** | **2020** | **23rd 2021** |
| **Household surveys (HHS)[a]** | | | | | |
| Tests used | Maglumi | Maglumi | Roche+Maglumi | Roche+Maglumi | Roche+Maglumi |
| All | 4.7 | 11.4 | 17.9 | 26.2 | 29.9 |
| | (2.7–7.0) | (9.2–13.6) | (15.0–20.9) | (22.5–29.9) | (26.2–33.5) |
| | (N = 517) | (N = 1183) | (N = 1470) | (N = 1129) | (N = 1194) |
| Wealthiest (Half) | NA | 6.5 | 9.4 | 21.6 | 22.8 |
| | | (4.4–8.5) | (6.0–12.8) | (17–26.3) | (18.1–27.6) |
| Poorest (Half) | | 16.0 | 22.0 | 30.4 | 36.4 |
| | | (12.2–19.8) | (17.2–26.7) | (24.7–36.0) | (31.1–41.8) |
| Women only | NA | 11.2 | 18.4 | 25.7 | 30.7 |
| | | (8.5–13.9) | (15.2–22.1) | (21.7–29.6) | (27.1–34.4) |
| | | (N = 632) | (N = 785) | (N = 603) | (N = 638) |
| **Women at delivery[b]** | | | | | |
| Wondfo | NA | 54.7 | 24.7 | 15.9 | 12.5 |
| | | (30.9–78.5) | (16.7–32.7) | (11.6–20.2) | (4.2–20.8) |
| Roche | NA | 44.1 | 32.7 | 27.5 | 28.0 |
| | | (21.8–66.4) | (24.1–41.4) | (22.3–32.7) | (16.4–39.5) |

[a] Project soroEpi MSP: https://monitoramentocovid19.org

[b] The percentages seropositive are from the spline fit through the whole data, rather than time- specific numerators and denominators.

## Discussion

The overall SARS-CoV-2 seroprevalence among pregnant women presenting at one of the largest university hospitals for delivery in Sao Paulo during the study period ranged from 17.9% to 28.9% depending on assay. The Roche Elecsys assay appeared more sensitive than Wondfo to identify participants with COVID-19 like symptoms or in the subgroup of women who were PCR-positive on nasopharyngeal swabs during pregnancy. Importantly, SARS-CoV-2 seroprevalence varied over time, reaching the highest levels at the beginning of the period (June 2020) and gradually decreasing, mirroring the COVID-19 pandemic's trajectory in São Paulo, when based on case reporting [3].

The presence of COVID-19 symptoms during pregnancy (particularly anosmia and ageusia in this early phase of the COVID-19 pandemic) was associated with SARS-CoV-2 seropositivity by either assay, highlighting the importance of eliciting symptoms through simple questionnaire in the diagnostic cascade. However, our data demonstrate that a relatively high proportion of asymptomatic women were also seropositive by Roche Elecsys (19%) or Wondfo (12%), underscoring the fact that without testing it would be challenging to estimate the actual size of the epidemic.

The choice of antibody assays is crucial for the quality of epidemiological information obtained. As pointed out by others, assays that detect and quantify long-lasting serological response are important to obtain reliable estimates of exposure. Conversely, immune assays with rapidly waning response may be useful for performing studies of recent infection [46]. We observed that the two assays used in this study behaved slightly differently, providing different profiles for antibody reactivity over time. The anti-SARS-CoV-2 total assay detecting

total antibodies against the spike S1 protein (Wondfo) is predicted to be highly sensitive to detect recent infection [45], and indeed a high prevalence was found in the early months of the epidemic, with good correlation with COVID-19 symptoms, but possibly waning thereafter in parallel with the actual COVID-19 epidemic, meaning it could be used as a diagnostic/confirmatory tool, but would be less accurate to determine the proportion infected up to that time. By contrast, the NC-directed antibody assay (Roche Elecsys) showed more steady signals over time, reflecting exposure during the different preceding periods, resulting in a possibly more accurate measure of cumulative risk [45].

As health authorities need to quickly know the proportion of the population infected or still susceptible to infection to guide public policies, serosurveillance in easily accessible, sensitive and, ideally, representative population groups is of paramount importance. This study shows that women presenting to delivery services, and, by extension, pregnant women could be a good sentinel population to track community infection spread for several reasons. Firstly, this group usually benefits from consecutive medical appointments during antenatal care, allowing for longitudinal monitoring at frequent intervals, but also represents a group that might still be exposed even at times of social isolation, as they need to at least come to delivery units, hence risk of infection can be measured. Secondly, in the early phase of the pandemic and prior to roll-out of a vaccination program, serology probably reflected well recent exposure, given the likely similar duration of pregnancy and half-life of SARS-CoV-2 antibodies [46]. Our study findings support these hypotheses. Three months into the pandemic, by June 2020, SARS-CoV-2 seroprevalence in women at delivery was notably higher than in the household survey population, which could be explained by the sustained exposure of pregnant women who had to travel to medical appointments during that period, whilst the general population had been requested to stay at home. From June 2020 on, with the official easing of social distancing [47], the difference in seroprevalence between the pregnant women and community populations (HHS) started to narrow, and seroprevalence was comparable thereafter. Infectious disease surveillance in pregnant women or women at delivery might thus be more sensitive than that obtained through household surveys because of the former population's greater exposure, more easily implemented and at lower cost. However, pregnant women are not representative of the whole female population because they tend to be younger, and, depending on recruitment sites, infection levels will reflect the particular socio-economic strata from their catchment area.

Other potential sentinel surveillance populations for infectious diseases include blood donors and healthcare workers. However, these two groups are also likely to be less homogeneously exposed than pregnant women. For example, data show that the attendance of blood donors had dropped significantly during lockdown measures [48–50], thus biasing estimates to lower levels. On the other hand, healthcare workers represent both a potentially more exposed population by nature of their work, but also a group more likely to receive and make proper use of personal protective equipment. Extrapolation of serological patterns obtained this group to the community's might be biased [51, 52].

The main limitations of our study include the lack of true longitudinal measurements of serological response among the same individuals to determine recency of infection through seroconversion, the lack of confirmation of true infection through molecular testing, particularly challenging to organize if a large proportion of infections may be pauci-symptomatic, and the relatively small sample size to determine associations with precision. These constraints are applicable to most serosurveillance surveys. Furthermore, an exact comparison of SARS-CoV-2 seropositivity between the HHS population and pregnant women was not possible, because seroprevalence in the former was based on positivity on either of the two assays (Maglumi and Roche). However, knowing that Roche was the most sensitive assay helps in the

interpretation of the HHS results, and we can compare trends over time between HHS and pregnant women.

As for other surveillance programs, the monitoring of infectious diseases trends is largely constrained by the choice of population, the performance and standardization of monitoring tools (ie, serological assays) and the trajectory of the epidemic/pandemic itself including response to it. In the case of SARS-CoV-2, appearance of new variants as well as the introduction of vaccination is prompting for adjustment of serological testing to be able to distinguish recent from past infection or vaccination and reflect the circulation of current variants.

In conclusion, we have shown that it was possible to organize SARS-CoV-2 serological surveillance among women coming to delivery units in Sao Paulo, that seroprevalence was both substantial and varying over time, reflecting the COVID-19 pandemic trajectory during that period. Pregnant women or women coming to delivery constitute a highly exposed and accessible population, suitable for the sentinel surveillance of emerging infections.

## Supporting information

**S1 Table. The STROBE statement- checklist of items that should be included in reports of cross-sectional studies.**
(DOCX)

**S2 Table. Factors associated with SARS-CoV-2 seropositivity by Wondfo.**
(DOCX)

## Acknowledgments

We acknowledge all members of the HC-FMUSP-Obstetric COVID-19 Study Group: Aline Scalisse Bassi; Amanda Wictky Fabri; Ana Claudia Rodrigues Lopes Amaral de Souza; Ana Claudia Silva Farche; Ana Maria Kondo Igai; Ana Maria da Silva Sousa Oliveira; Adriana Lippi Waissman; Carlos Eduardo do Nascimento Martins; Cristiane de Freitas Paganoti; Danielle Rodrigues Domingues; Fernanda Cristina Ferreira Mikami; Fernanda Spadotto Baptista; Jacqueline Kobayashi Cippiciani; Jéssica Gorrão Lopes Albertini; Joelma Queiroz de Andrade; Juliana Ikeda Niigaki, Marco Aurélio Knippel Galletta; Mariana Yumi Miyadahira, Mariana Vieira Barbosa; Monica Fairbanks de Barros; Nilton Hideto Takiuti; Sckarlet Ernandes Biancolin Garavazzo; Silvio Martinelli; Tiago Pedromonico Arrym; Ursula Trovato Gomez; Veridiana Freire Franco.

## Author Contributions

**Conceptualization:** Maria de Lourdes Brizot, Rossana Pulcineli Vieira Francisco, Philippe Mayaud.

**Formal analysis:** Neal Alexander.

**Funding acquisition:** Maria de Lourdes Brizot, Ester Cerdeira Sabino, Rossana Pulcineli Vieira Francisco, Philippe Mayaud.

**Investigation:** Mariana Yumi Miyadahira, Mara Sandra Hoshida, Ana Maria da Silva Sousa Oliveira, Ana Claudia Silva Farche.

**Methodology:** Mariana Yumi Miyadahira, Maria de Lourdes Brizot, Neal Alexander, Philippe Mayaud.

**Project administration:** Maria de Lourdes Brizot, Rossana Pulcineli Vieira Francisco.

**Resources:** Ester Cerdeira Sabino, Lea Campos de Oliveira da Silva.

**Supervision:** Maria de Lourdes Brizot.

**Writing – original draft:** Mariana Yumi Miyadahira, Maria de Lourdes Brizot, Neal Alexander, Ester Cerdeira Sabino, Philippe Mayaud.

**Writing – review & editing:** Mariana Yumi Miyadahira, Maria de Lourdes Brizot, Neal Alexander, Rossana Pulcineli Vieira Francisco, Philippe Mayaud.

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
