## [Decision Letter · Decision Letter 0]

11 Aug 2022

PONE-D-22-16593

Monitoring SARS-CoV-2 seroprevalence over time among pregnant women admitted to delivery units: suitability for surveillance

PLOS ONE

Dear Dr. MAYAUD,

Thank you for submitting your manuscript to PLOS ONE. After careful consideration, we feel that it has merit but does not fully meet PLOS ONE’s publication criteria as it currently stands. Therefore, we invite you to submit a revised version of the manuscript that addresses the points raised during the review process.

The reviewer has recommended that you make minor revisions to the manuscript. Please attend to their concerns and return the revised manuscript as advised in this letter.

We look forward to receiving your revised manuscript.

Kind regards,

Martin Chtolongo Simuunza, PhD

Academic Editor

PLOS ONE

Journal Requirements:

a) Did participants provide their written or verbal informed consent to participate in this study?

4. Please upload a copy of Supporting Information Table 1 which you refer to in your text on page 8.

5. Please upload a copy of Supporting Information Table 2 which you refer to in your text on page 14.

Reviewers' comments:

Reviewer's Responses to Questions

**Comments to the Author**

1. Is the manuscript technically sound, and do the data support the conclusions?

Reviewer #1: Yes

2. Has the statistical analysis been performed appropriately and rigorously? 

Reviewer #1: Yes

3. Have the authors made all data underlying the findings in their manuscript fully available?

Reviewer #1: Yes

4. Is the manuscript presented in an intelligible fashion and written in standard English?

Reviewer #1: Yes

5. Review Comments to the Author

Reviewer #1: In the manuscript "Monitoring SARS-CoV-2 seroprevalence over time among pregnant women admitted to delivery units: suitability for surveillance" the authors propose that pregnant women could constitute a population suitable for sentinel surveillance not only for COVID-19 but for other emerging infectious diseases aw well.

My main concern is that the laboratory methods used for the antibody determination in the HHS and the study group are different with different sensitivity and specifity, in order to compare the percentages. I think that at least it should be stated in the limitations of the study.

Otherwise, the data supporting this hypothesis are sufficient and the statistcal analysis has been performed appropriately.

6. PLOS authors have the option to publish the peer review history of their article (what does this mean?). If published, this will include your full peer review and any attached files.

Reviewer #1: No

---

## [Author Response · Author response to Decision Letter 0]

22 Nov 2022

RESPONSE TO THE EDITOR

Reply: We have reviewed PLOS ONE’s style requirements and made the adjustments to the text. 

a) Did participants provide their written or verbal informed consent to participate in this study?

Reply: In the “Ethical considerations” (within the Materials and methods section), at p.10 it is stated that: “All participants provided signed informed consent.” Also, in the “Settings and population” (within the Materials and methods section) it is explained that: “Those pregnant women of all ages who signed the informed consent form to participate in the study were eligible and included. “

3. We note that you have indicated that data from this study are available upon request. PLOS only allows data to be available upon request if there are legal or ethical restrictions on sharing data publicly. 

Reply: We have uploaded the data set to the OSF repository and the DOI is:10.17605/OSF.IO/8QHZP

4. Please upload a copy of Supporting Information Table 1 which you refer to in your text on page 8.

Reply: We have now uploaded Supporting Information Table 1. 

5. Please upload a copy of Supporting Information Table 2 which you refer to in your text on page 14.

Reply: We have now uploaded Supporting Information Table 2. 

Reply: We have noticed that some webpages we have cited were not available anymore and that the citations needed adjustments. So, we have made corrections to the following citations:

1. WHO Coronavirus (COVID-19) Dashboard. 2020. Available from: https://covid19.who.int.

2. Painel Coronavírus. 2022. Available from: https://covid.saude.gov.br.

3. Boletim diário COVID-19. 2020 .Available from: https://www.prefeitura.sp.gov.br/cidade/secretarias/upload/saude/20220531_boletim_covid19_diario.pdf

32. Interim Guidelines for COVID-19 Antibody Testing. 2020. Available from: https://www.cdc.gov/coronavirus/2019-ncov/lab/resources/antibody-tests-guidelines.html.

35. SoroEpi MSP. 2020. Available from: https://www.monitoramentocovid19.org.

37. Coronavirus disease (COVID-19)- Symptoms. 2020. Available from: https://www.who.int/health-topics/coronavirus#tab=tab_3.

47. Diário Oficial- Estado de São Paulo. 2020. Available from: http://diariooficial.imprensaoficial.com.br/nav_v5/index.asp?c=4&e=20200529&p=1.

RESPONSE TO REVIEWER

Reviewer Comments:

Comments to the Author

Reviewer #1: In the manuscript "Monitoring SARS-CoV-2 seroprevalence over time among pregnant women admitted to delivery units: suitability for surveillance" the authors propose that pregnant women could constitute a population suitable for sentinel surveillance not only for COVID-19 but for other emerging infectious diseases as well.

My main concern is that the laboratory methods used for the antibody determination in the HHS and the study group are different with different sensitivity and specifity, in order to compare the percentages. I think that at least it should be stated in the limitations of the study.

Otherwise, the data supporting this hypothesis are sufficient and the statistcal analysis has been performed appropriately.

Reply: We have now added 2 sentences in the DISCUSSION section (p.22-23) about this issue: 

…” Furthermore, an exact comparison of SARS-CoV-2 seropositivity between the HHS and pregnant women was not possible, because seroprevalence in the former was based on positivity on either of the two assays (Maglumi and Roche). However, knowing that Roche was the most sensitive assay helps in the interpretation of the HHS results, and we can compare trends over time between HHS and pregnant women. “

---

## [Decision Letter · Decision Letter 1]

20 Dec 2022

Monitoring SARS-CoV-2 seroprevalence over time among pregnant women admitted to delivery units: suitability for surveillance

PONE-D-22-16593R1

Dear Dr. MAYAUD,

We’re pleased to inform you that your manuscript has been judged scientifically suitable for publication and will be formally accepted for publication once it meets all outstanding technical requirements.

Kind regards,

Martin Chtolongo Simuunza, PhD

Academic Editor

PLOS ONE

Additional Editor Comments (optional):

Reviewers' comments:

Reviewer's Responses to Questions

**Comments to the Author**

1. If the authors have adequately addressed your comments raised in a previous round of review and you feel that this manuscript is now acceptable for publication, you may indicate that here to bypass the “Comments to the Author” section, enter your conflict of interest statement in the “Confidential to Editor” section, and submit your "Accept" recommendation.

Reviewer #1: All comments have been addressed

2. Is the manuscript technically sound, and do the data support the conclusions?

Reviewer #1: Yes

3. Has the statistical analysis been performed appropriately and rigorously? 

Reviewer #1: I Don't Know

4. Have the authors made all data underlying the findings in their manuscript fully available?

Reviewer #1: Yes

5. Is the manuscript presented in an intelligible fashion and written in standard English?

Reviewer #1: Yes

6. Review Comments to the Author

Reviewer #1: (No Response)

7. PLOS authors have the option to publish the peer review history of their article (what does this mean?). If published, this will include your full peer review and any attached files.

Reviewer #1: No

---

## [Editor Report · Acceptance letter]

26 Dec 2022

PONE-D-22-16593R1 

Monitoring SARS-CoV-2 seroprevalence over time among pregnant women admitted to delivery units: suitability for surveillance 

Dear Dr. Mayaud:

I'm pleased to inform you that your manuscript has been deemed suitable for publication in PLOS ONE. Congratulations! Your manuscript is now with our production department. 

Kind regards, 

on behalf of

Dr. Martin Chtolongo Simuunza 

Academic Editor

PLOS ONE